# A Bibliometric Analysis of the Scientific Outcomes of European Projects on the Digital Transformation of SMEs

Fernando Almeida [1,*] , José Morais [2,3] and José Duarte Santos [3]

1 Faculty of Engineering, University of Porto and INESC TEC, 4200-465 Porto, Portugal
2 Polytechnic Higher Institute of Gaya (ISPGAYA) and CEOS.PP, 4400-103 Vila Nova de Gaia, Portugal
3 Accounting and Business School of the Polytechnic of Porto (ISCAP/P.PORTO) and CEOS.PP, 4465-004 Porto, Portugal
* Correspondence: almd@fe.up.pt

**Abstract:** The projects funded under the European Horizon 2020 program have responded to the challenges facing small enterprises and have provided a framework for different actors (e.g., universities, R&D centers, SMEs) to collaborate and find innovative approaches to address the challenges of digital transformation. This study conducts a bibliometric analysis of the scientific production supported by this project, between 2014 and 2021, evaluating 114 projects, which have associated 2312 scientific production items and 1460 deliverables. The results demonstrate that scientific production is mostly carried out collaboratively with project partners and is mainly published in peer-reviewed journals. The research demonstrates that resources, such as Horizon 2020, provide a useful adjunct to other databases as a basis for bibliometric and related analyses.

**Keywords:** EU; Horizon 2020; bibliometric analysis; SMEs

## 1. Introduction

Research and development (R&D) plays a key role in stimulating sustainable growth and job creation [1–4]. In the European context, this phenomenon has been widely recognized, as demonstrated by the studies carried out by Beugelsdijk et al. [5] and Sanyé-Mengual et al. [6] that highlight the role of productivity in driving sustainable economic growth and consolidating the European recovery. Furthermore, R&D investments are central to the development of new products, services and processes that increase productivity and sectoral competitiveness [7–11].

R&D is a collaborative activity, and the emergence of innovation networks demonstrates the importance of different players from the public and private sectors, from education and industry, working together to address the great challenges of society [12,13]. The results of the study by Fu et al. [14] show that research collaboration across multiple institutions, countries and regions increases the quality, visibility and impact of scientific production. The European Union has responded to this challenge through multiple Framework Programs (FPs) that have been created since 1984, and since that year, they have fostered the emergence of inter-organizational and inter-regional cooperation projects [15]. These programs have increased the competitiveness of the research generated and reduced the technological gap with the USA [16].

The sequential numbering of the various European support programs for research and innovation was discontinued in 2014, with the emergence of Horizon 2020. This was the support program for the period 2014–2020 and had an overall budget of more than EUR 77 billion to support research, technological development, demonstration and innovation projects [17]. Horizon 2020 was based on three main pillars of action: (i) scientific excellence, which seeks to strengthen and extend the excellence of scientific production making it more competitive on a global scale; (ii) industrial leadership, which aims to accelerate the

development of technologies and innovations in companies and help small and medium-sized enterprises (SMEs) to grow and become world leaders; and (iii) societal challenges, which take on the challenge of supporting activities from research to market [17]. Funding covers multiple areas of society from health, agriculture, forestry, environment, energy efficiency, sustainability, etc.

SMEs play a central role in increasing the competitiveness of the European Union countries and are, therefore, a central element of the Horizon 2020 policy. Statistics show that there will be about 22.6 million SMEs in the European Union in 2021, which is about 99% of all businesses [18]. Turning SMEs more competitive on a global scale is part of Horizon 2020 and digitalization is a path that SMEs must follow to achieve this goal. Studies conducted by Bouwman et al. [19] and Trischler and Li-Ying [20] show that digitalization has an immediate and disruptive effect on SMEs' business models. Moreover, digitalization also contributes to innovation in more traditional businesses that generate many efficiencies in value chains [21]. In this sense, SMEs should look at digital transformation processes beyond adapting processes and business areas to new technologies and instead as a way to find new solutions that can be solved with the help of technology.

Exploring and measuring the role of these digital transformation projects in SMEs is relevant to understanding the impact of programs supporting research and innovation. Since its formation in 2014, Horizon 2020 has established a set of key performance indicators (KPIs) to assess the performance of this initiative. A total of 23 indicators are defined and include areas, such as scientific production, knowledge transfer, and social impact of projects [22]. In the sub-area of scientific production, there are only two indicators specific to scientific production that tend to measure the number of publications in peer-reviewed high-impact journals and the percentage of publications resulting from projects funded among the top 1% highly cited. Despite the importance of these indicators for analyzing the relevance of scientific production, they do not offer a level of granularity fine enough to explore in-depth the phenomenon of scientific production considering different types of publications and the degree of collaboration among the various entities participating in each project. Our study addresses this research gap by carrying out a bibliometric study of the scientific results produced by projects funded by Horizon 2020, specifically aimed at supporting the digitalization processes of SMEs.

This study is organized as follows: In the first phase, the research questions that frame this bibliometric study are defined and substantiated. After that, the methodology of the study with its various phases and methods is presented. Next, the results of the study are explored and discussed considering their relevance in the context of each research question. Finally, the theoretical and practical contributions of this study are explored. The main limitations of the study are also briefly reviewed and suggestions for future work are provided.

## 2. Research Questions

The visibility on the status of each project supported by the Horizon 2020 program is achieved through two instruments: deliverables and milestones. Deliverables are outputs of a very diverse nature, such as prototypes, demonstrators, technical reports, brochures, scientific publications, etc. In turn, milestones are control points in the project that help to trace and follow the evolution of projects, which include the final delivery point of the project, but also its intermediate points [23]. Therefore, scientific production is not the only scientific outcome of a project, and it is relevant to explore the relative weight of scientific production, compared with other types of evidence. Therefore, the first research question of this study was defined:

*RQ1. What is the relative weight of scientific production in the project outcomes?*

The bibliometric analysis of the scientific production in the digital economy and technological transformation of organizations has been a topic that has aroused strong interest in the scientific community in recent years. We highlight the works carried out by Borregan-Alvarado et al. [24] in the industry 4.0 and advanced manufacturing, Montalván-

Burbano et al. [25] in organizational innovation, and Pan et al. [26] in the digitalization of the economy. The methodological approach of these studies presents several common points, such as the analysis of the distribution of publications by year and typology. In this sense, this study follows a similar approach in addressing both research questions:

*RQ2. What is the distribution of publications per year?*

*RQ3. What is the distribution of publications by typology?*

Systematic reviews also stand out by identifying the number of cations per author and per document. This is an approach that has been adopted in studies in the field of digital transformation [27,28]. In this sense, the following research question was defined:

*RQ4. Which are the most cited authors and documents?*

The role of SMEs over the years has been widely recognized in the European Union [29–35]. However, their size makes them particularly vulnerable to external changes. Added to this is their specific positioning and challenges within each sector of activity. In brief, some challenges are transversal to SMEs and others arise specifically connected to the sector of activity. In particular, digital transformation has a set of barriers that emerge in specific contexts of activity sectors [36,37]. Accordingly, a new research question has been established:

*RQ5. What are the sectors of activity of SMEs addressed by the publications?*

An essential feature of the projects funded by Horizon 2020 is the establishment of a network of partners. Horizon 2020 provides tools for a choice of partners depending on the thematic areas of projects [38]. Consequently, it is expected that the scientific production reflects the collaborative research work of the partners. Accordingly, the following research question was defined:

*RQ6. What is the percentage of international collaborations per publication?*

It has been increasingly recognized that open access is a key element in the advancement of science. Open access is based on the premise that scientific knowledge is a public good and, therefore, should be available to all. It is an alternative to the traditional publishing model that restricts access to content through paid subscriptions. The literature exposes a significant set of benefits of the open access model, such as increased visibility of research results, maximization of the potential for international collaboration of research activities, and increased citation potential [39–41]. In this sense, a research question was defined that intends to explore this phenomenon:

*RQ7. What is the percentage of open access publications per R&D project?*

The financial support that is granted to each project is not equal and is allocated based on competitive calls and through an independent evaluation process of the proposals submitted. The budget is distributed over three pillars: pillar I of scientific excellence with about 32% of the budget; pillar II of industrial leadership, which accounts for about 22% of the budget; and pillar III of societal challenges with about 39% of the budget [42]. In addition to the themes covered by the three pillars, other projects may be funded under other instruments with a total of 7% of the budget. The number of partners in each project is quite diversified. Horizon 2020 stipulates, in the minimum conditions for participation, that three independent legal entities from three different member states or associated states are required, which means that more than one entity per country may participate if the minimum is assured [43]. Therefore, it is relevant to explore the scientific production considering the budget of the project and the number of partners involved. To this end, two more research questions were defined:

*RQ8. What is the correlation of R&D project amount funding with the number of publications?*

*RQ9. What is the correlation of the number of partners in the R&D project with the number of publications?*

## 3. Materials and Methods

According to Donthu et al. [44], bibliometric studies adopt quantitative and statistical techniques to measure the rates of production and dissemination of scientific knowledge. In this sense, this technique allows for identifying and measuring patterns of written

communication, as well as the authors of these communications. This approach has become popular among the scientific community due to the large amount of biographical material that is currently produced and made available. Having a summarized and systematized view of this can facilitate understanding and even point to future research paths. Therefore, the results of a bibliometric study can help young researchers or even the more experienced ones who come across a new theme.

Figure 1 presents the four phases of the methodological process. The data were collected, compiled and analyzed between 18 April–15 June 2022. The first methodological phase starts with database selection. CORDIS is the official European Commission (EC) portal for planning, applying and monitoring the status of projects in the Horizon 2020 program. The CORDIS portal was chosen because it is publicly accessible and has reliable information about the scientific production and technical deliverables associated with each project funded by Horizon 2020. WoS or Scopus databases could not be adopted because the search in these databases is performed based on search terms and looks exclusively at scientific production. Other approaches have been adopted to collect information on scientific productivity [45]. CORDIS offers a completely different approach. It is not a scientific database but a database of European projects, in which the scientific output is associated with each project. CORDIS database has been used as a source of information for collecting data on projects funded by the European Union [46–48]. The following information is available for each project in CORDIS: (i) project identifier code; (ii) project acronym; (iii) project title; (iv) project status; (v) funding program; (vi) start and end date; (vii) project goal; (viii) project cost and maximum funding contribution; (ix) funding scheme; (x) project participants; and (xi) project coordinator [49]. The second phase is aimed at selecting the search criteria. Three attributes were established: (i) inclusion in the Horizon 2020 (H2020) program; (ii) application domain must focus on the digital economy among SMEs; and (iii) project completion date by the end of 2021. Therefore, all projects that have not been completed by 2021 have been excluded. CORDIS allows selecting these search parameters on the online platform, except for the focus on SMEs, whose search had to be based on a manual inspection. A total of 114 projects were found to meet these criteria. Stage 3 looks to the software that was used for data exploration. Microsoft Excel was used for descriptive statistical analysis of the data and graph construction, SPSS for correlational analysis, and the VOSViewer for bibliometric network analysis and mapping. Finally, the last phase considers a different set of techniques that were used to answer the previously selected research questions. At this level, performance analysis and science mapping were performed with the help of the VOSViewer, which is a tool for building and visualizing bibliometric networks. This software has been used in several biometric studies in areas, such as social cohesion, public networks and circular economy [50–57].

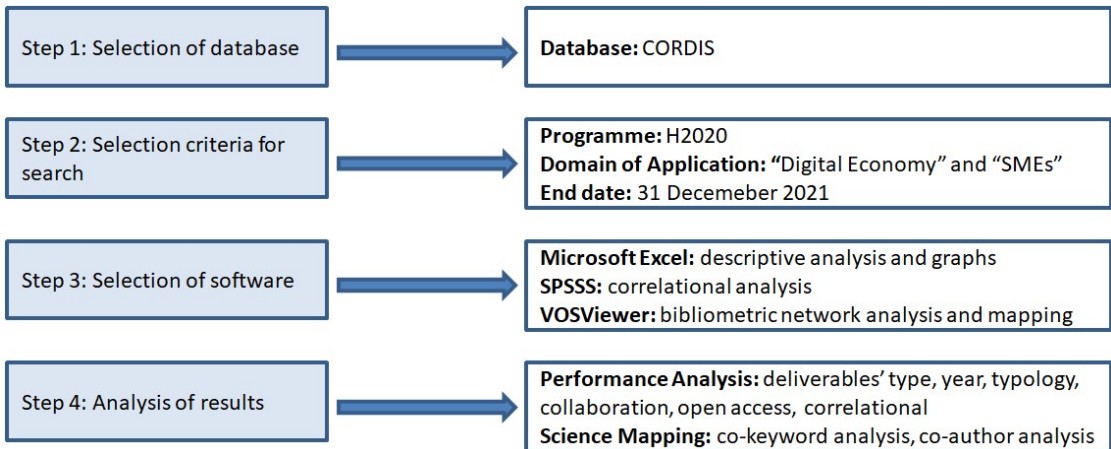

**Figure 1.** Phases of the methodology.

The activity sector addressed by each publication considers the benchmark of the International Standard Industrial Classification of All Economic Activities (ISIC) in revision 4.0. Therefore, a total of 20 activity sectors plus 1 unclassified sector as proposed in ISIC framework [58] are included in this study (Figure 2). An automatic inspection of the title and abstract of each publication was performed in search of the keywords associated with each ISIC activity sector using the VOSViewer.

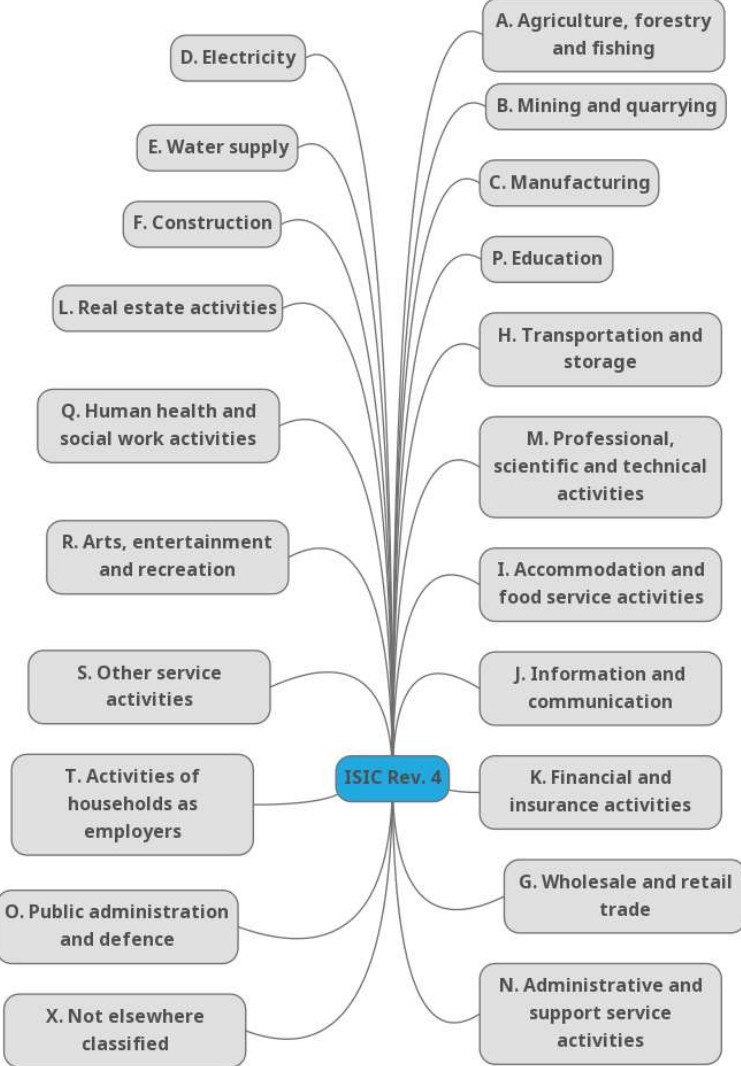

**Figure 2.** Sectors of activity represented in ISIC.

## 4. Results

### 4.1. RQ1. What Is the Relative Weight of Scientific Production in the Project Outcomes?

Table 1 presents the projects included in the listing ordered decreasingly by the total number of publications. The relative weight of publications for each project is also calculated. Only projects with more than 100 items of scientific production are presented. The remaining projects are grouped in an "other" class. In total, considering all projects, 2312 scientific output items and 1460 deliverables were published. Only in two of the seven projects with the highest scientific production is the relative weight of scientific production below 85%. The scientific production focuses on a smaller number of projects while the deliverables show lower levels of concentration. Indeed, in the projects with lower scientific production, the deliverables represent a higher percentage of the outcomes of these projects.

**Table 1.** Project outcomes.

| ID | Scientific Production | Deliverables | Relative Weight |
|---|---|---|---|
| ACTRIS-2 | 441 | 74 | 0.8563 |
| Construye2020_Plus | 186 | 18 | 0.9118 |
| AtlantOS | 177 | 111 | 0.6146 |
| ECOPOTENTIAL | 174 | 59 | 0.7468 |
| IPERION CH | 122 | 13 | 0.9037 |
| Smart Exploration | 108 | 13 | 0.8926 |
| ANYWHERE | 101 | 16 | 0.8632 |
| "Other" | 1003 | 1156 | 0.4646 |

*4.2. RQ2. What Is the Distribution of Publications per Year?*

Figure 3 shows the distribution of scientific production related to the projects funded by Horizon 2020. The starting year of this European funding program was considered to be 2014 and an additional year of analysis was added, given the period of operation of this program (year of 2021), taking into account that part of the scientific production of the projects can be carried out after their conclusion. The data show that 81.62% of the scientific production was carried out in the four central years of this program (between 2016 and 2019). On the contrary, scientific production in 2014 is quite residual (1.30%), which reveals the difficulties of showing evidence of scientific production in the first years of launching a program of funding for scientific production. On the opposite side, and after the deadline for completion of projects, we found 115 publications (4.97%) relative to 2021.

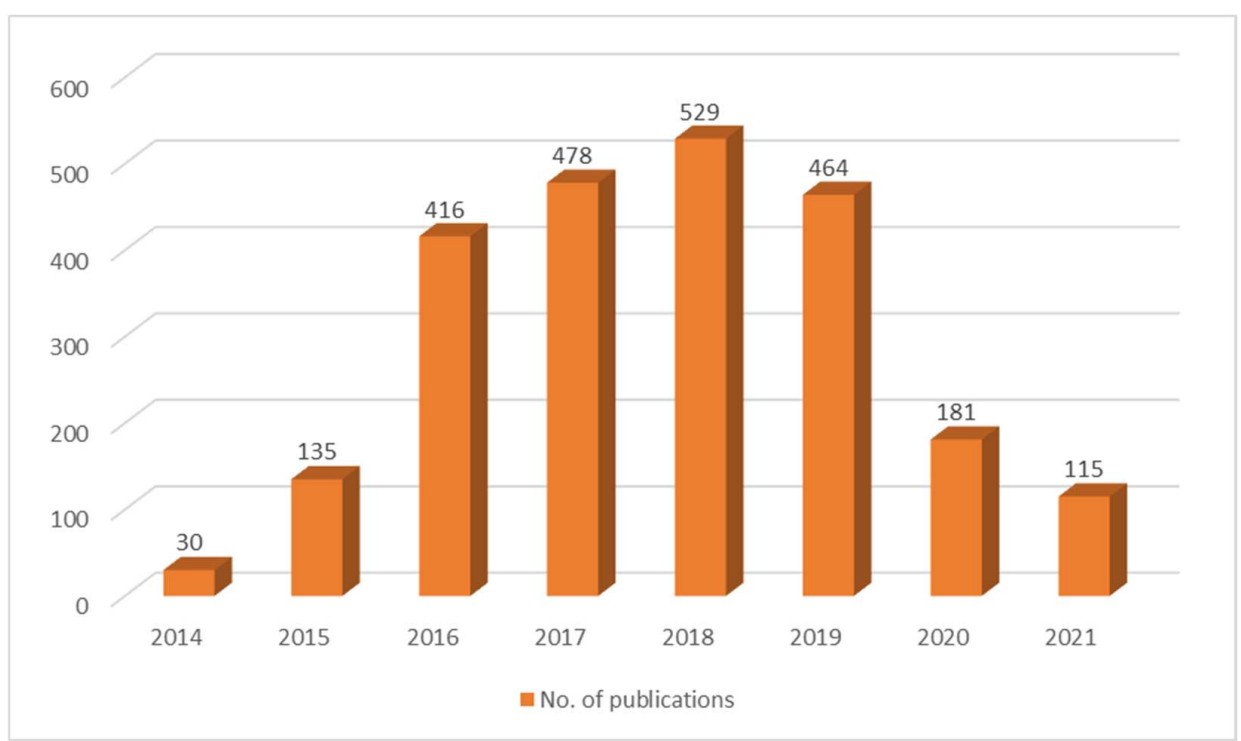

**Figure 3.** Number of publications per year.

*4.3. RQ3. What Is the Distribution of Publications by Typology?*

Figure 4 shows the distribution of project outcomes considering the deliverables (represented in the figure with green tones) and scientific production (represented in the figure with blue tones). The projects present a total of 1460 deliverables and 2348 publications. It is graphically shown that the projects tend to present both outcomes, although there is a predominance of the outcomes related to the scientific production which represents 61.66% of the evidence of the projects. The largest percentage of scientific production is related to

peer reviewed articles (*n* = 1092) that represent 47.23% of the scientific production of the projects. In contrast, book chapters (*n* = 61) represent only 2.60%.

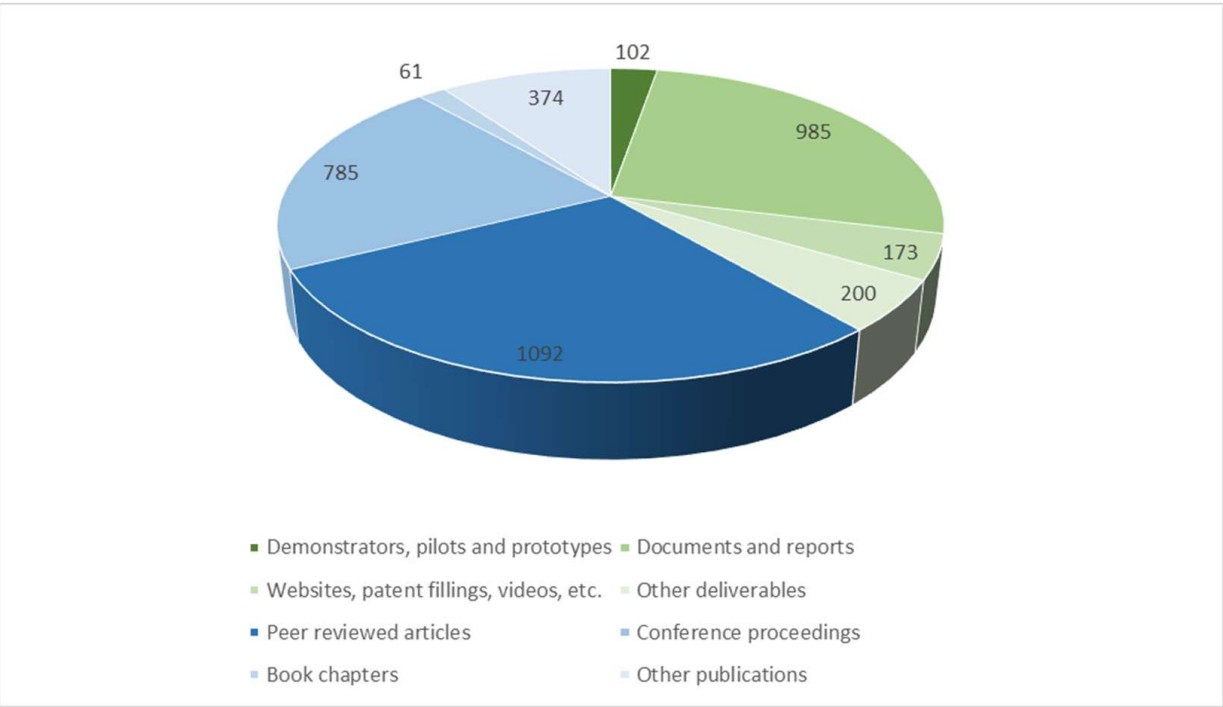

**Figure 4.** Project deliverables and scientific production.

*4.4. RQ4: Which Are the Most Cited Authors and Documents?*

Table 2 presents the top 10 most cited authors. In addition to the total number of citations per author, the number of documents written by each author is also presented. It is noted that six of these authors have published nine or more studies. In total, the 10 most cited authors represent 4961 citations.

**Table 2.** Top 10 most-cited authors.

| Author | Documents | Citations |
|---|---|---|
| Aghakouchak, Amir | 3 | 774 |
| Ward, Philip J. | 4 | 679 |
| Kulmala, Markku | 9 | 539 |
| Petäjä, Tuukka | 9 | 513 |
| Fraaije, Marco W. | 13 | 500 |
| Ansmann, Albert | 12 | 421 |
| Kerminen, Veli-Matti | 4 | 421 |
| Baars, Holger | 12 | 384 |
| Provenzale, Antonello | 9 | 365 |
| Turco, Marco | 5 | 365 |

Table 3 presents the top 10 most cited publications. For each publication, the corresponding author, the year, the title, and the number of citations are presented. It is noted that 240 publications have 25 or more citations. In the top-10 most-cited publications there are 3070 citations. It is also noted that each of these publications has more than 200 citations.

**Table 3.** Top 10 most-cited documents.

| Correspondence Author | Year | Title | Citations |
|---|---|---|---|
| Zscheischler, Jakob | 2018 | Future climate risk from compound events | 598 |
| Le Quéré, Corinne | 2015 | Global Carbon Budget 2015 | 499 |
| Olsen, Are | 2016 | The Global Ocean Data Analysis Project version 2 (GLODAPv2)—an internally consistent data product for the world ocean | 309 |
| Dudas, Gytis | 2017 | Virus genomes reveal factors that spread and sustained the Ebola epidemic | 275 |
| Bielejec, Filip | 2016 | SpreaD3: Interactive Visualization of Spatiotemporal History and Trait Evolutionary Processes | 271 |
| Bianchi, Federico | 2016 | New particle formation in the free troposphere: A question of chemistry and timing | 266 |
| Goodenough, Kathryn | 2018 | The Rare Earth Elements: Demand, Global Resources, and Challenges for Resourcing Future Generations | 230 |
| Firtz, Steffen | 2019 | Citizen science and the United Nations Sustainable Development Goals | 214 |
| Romero, Elvira | 2018 | Same Substrate, Many Reactions: Oxygen Activation in Flavoenzymes | 206 |
| Kerminen, Veli-Matti | 2018 | Atmospheric new particle formation and growth: review of field observations | 202 |

*4.5. RQ5. What Are the Sectors of Activity of SMEs Addressed by the Publications?*

Initially, a bibliometric network of the projects' publications was constructed using the VOSViewer software. Each publication was imported into the software using its digital object identifier (DOI). The network was built considering the title and abstract fields. Structured abstract labels and copyright status were ignored. The software provides two counting models: (i) complete counting means that all occurrences of a term in a document are counted; and (ii) binary counting means that only the presence or absence of a term in a document matters, which means that the number of occurrences of a term in a document is not taken into account. In this study, it is considered relevant to count the absolute number of occurrences of the terms in the documents. Stop words (e.g., "and", "the", "one", "sup", "sub", "after", etc.) were eliminated from the network construction process because they were considered irrelevant. Moreover, all terms that do not fit within a digital transformation application area (e.g., "task", "goal", "paper", "solutions", "company", "forecast", "goal", etc.) have also been removed. Therefore, it is identified, mapped and clustered the most relevant terms as referred in Waltman et al. [59]. In the network construction, a minimum number of occurrences of a term equal to three was set. Of the 9584 terms, 1065 meet the threshold. For each of the 1065 terms, a relevance score was calculated, and 30 terms were included in the final network, as represented in Figure 5. The terms "paper", "food security", "water", "ecosystem system", "cultural heritage" and "agriculture" stand out as the main terms in the studies. This process of identifying the most relevant terms associated with a given knowledge field or the search for an association between publications and citations has been adopted in bibliometric studies [60–62]. Furthermore, the adoption of the VOSViewer also allowed us to identify the emergence of seven clusters, as shown in Table 4.

After that, the identified terms were mapped in the ISIC framework to find the associated activity sectors in each study. Only the title and abstract of each study were analyzed. A dictionary of synonyms of terms related to each activity sector was also built to allow the aggregate counting of studies that focus on the same activity sector, despite the term not being exactly the same or representing a subsector of activity (e.g., manufacturing, production, furniture, textiles, chemicals, pharmaceuticals, etc.). In the end, the treemap in Figure 6 was built, which visually represents the relative weight of the sectors of activity. It is noted that 65.7% of the projects have applicability in at least one activity sector, which indicates that most European projects have a high focus on applied research, to the detriment of fundamental research. Most of the publications are in multidimensional sustainability considering the thematic domains related to the environment, technologies and people. Water supply is the activity sector with the highest number of publications

(*n* = 354), followed by the sectors related to information and communication (*n* = 284) and human health and social work activities (*n* = 212).

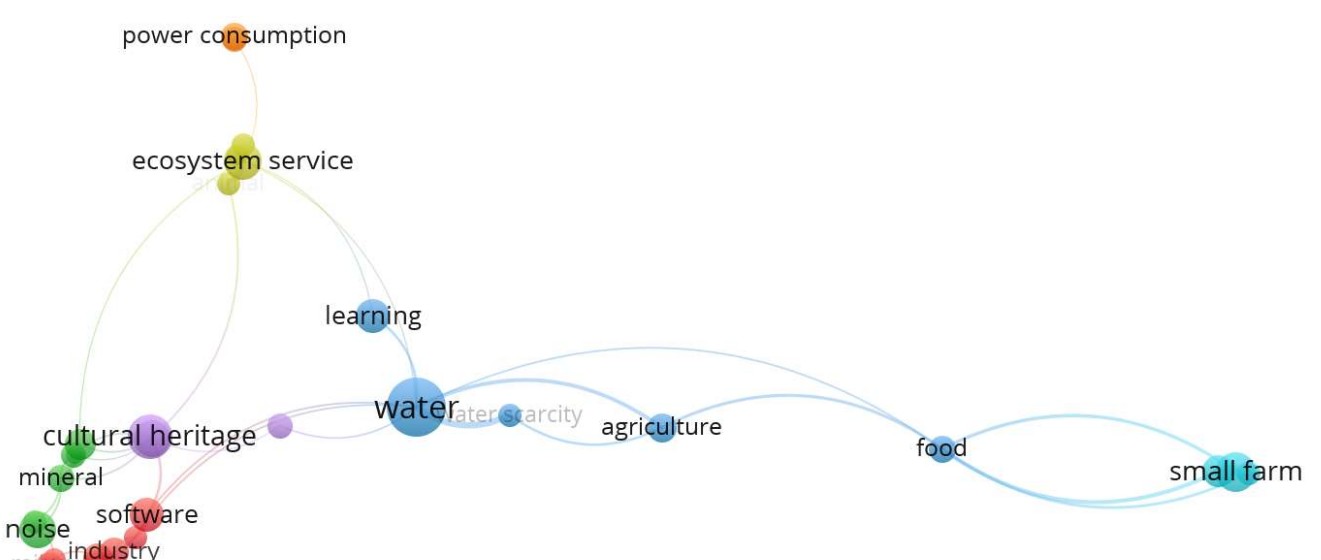

**Figure 5.** Network visualization of application fields.

**Table 4.** Grouping the key terms into clusters.

| Cluster | Key Terms |
|---|---|
| A | "earthquake", "hardware", "industry", "mineral exploration", "railway", "software" and "underground mine" |
| B | "air quality", "ambient noise", "black carbon", "mineral" and "noise" |
| C | "agriculture", "food", "learning", "marine", "underwater environment", "water" and "water scarcity" |
| D | "animal", "citizen science", "ecosystem service" and "wildfire" |
| E | "cultural heritage" and "reconstruction" |
| F | "farm", "food security" and "small farm" |
| G | "earth observation" and "power consumption" |

*4.6. RQ6. What Is the Percentage of International Collaborations per Publication?*

Table 5 presents the distribution of the frequency and percentage of international collaborations in publications in the context of the R&D projects. The data have been aggregated considering non-uniform value ranges to facilitate better data analysis. Most of the projects have a collaboration rate equal to 100% (i.e., all publications include international partners) and more than 68% of the projects have a collaboration rate higher than 90%. On the contrary, less than 10% of the projects have an international collaboration rate below 75% and only one project has a collaboration rate below 50%. The only project in this situation is "Construye2020_Plus", which presents very specific particularities, since 181 of the 186 scientific artifacts produced in this project correspond to news in the local press promoted individually by each of the six project partners.

**Table 5.** International collaboration.

| Interval | Frequency | Percent | Cumulative Percent |
|---|---|---|---|
| [0–50] | 1 | 0.0196 | 0.0196 |
| [50–75] | 4 | 0.0784 | 0.0980 |
| [75–90] | 11 | 0.2157 | 0.3137 |
| [90–100] | 13 | 0.2549 | 0.5686 |
| Equal to 100 | 22 | 0.4314 | 1 |

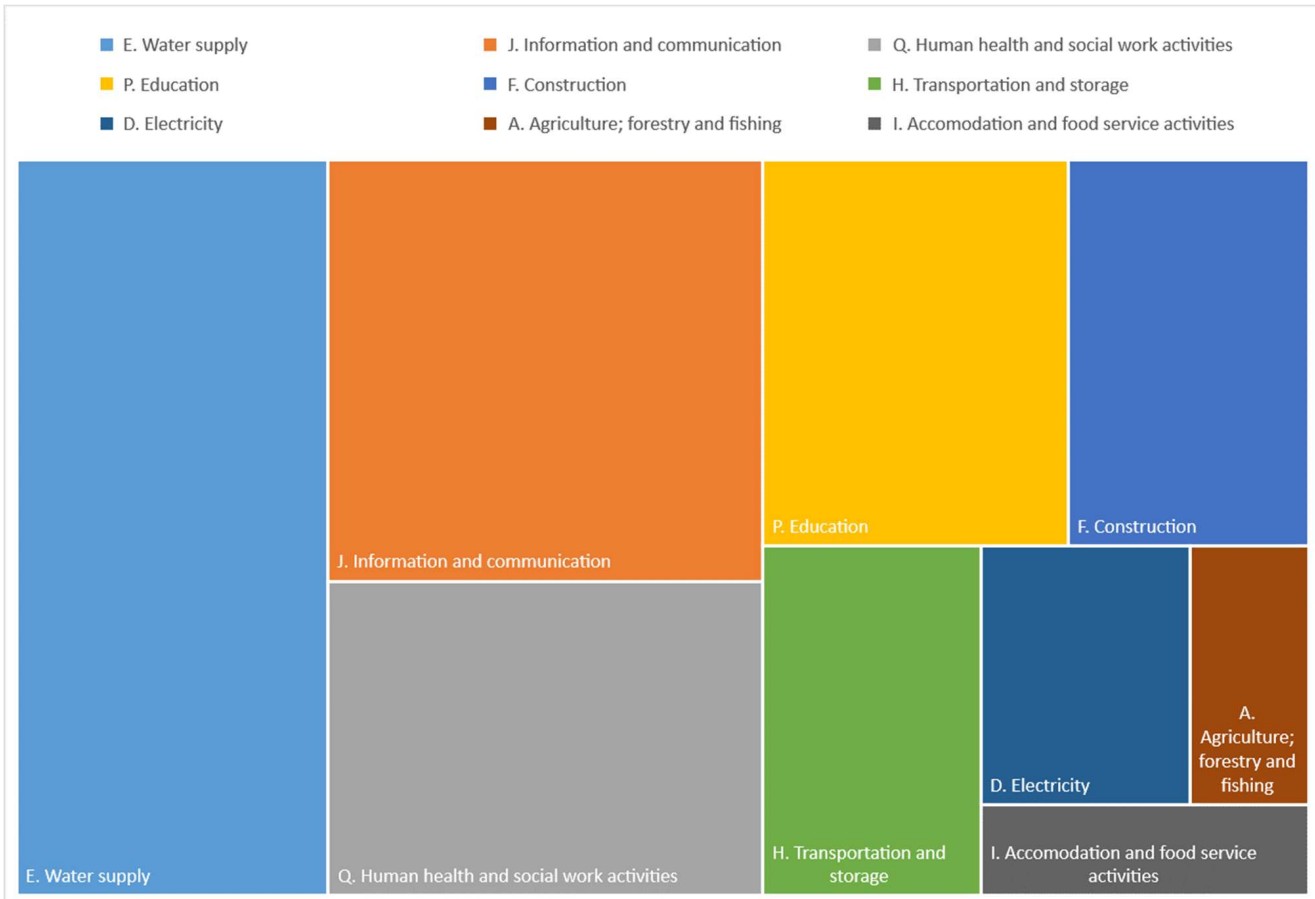

**Figure 6.** Treemap of the relative weight of the sectors of activity.

### 4.7. RQ7. What Is the Percentage of Open Access Publications per R&D Project?

Table 6 presents the distribution of publications published in open access in the context of R&D projects. The data were also aggregated in classes, but homogeneous intervals were considered, since there are significant oscillations among the projects. The last interval also includes the value 100. The largest number of open access publications is found in the interval from 60 to 70, which represents close to 25% of the total scientific production. In the middle intervals (i.e., from 20 to 70), more than 80% of the scientific production is found. Two projects have an open access publication rate higher than 90%, respectively: "Construye2020_Plus" with 90.06% and "WISER" with 100%.

**Table 6.** Open access publications.

| Interval | Frequency | Percent | Cumulative Percent |
| --- | --- | --- | --- |
| [0–10] | 6 | 0.1176 | 0.1176 |
| [10–20] | 0 | 0 | 0.1176 |
| [20–30] | 5 | 0.0980 | 0.2156 |
| [30–40] | 9 | 0.1765 | 0.3921 |
| [40–50] | 7 | 0.1373 | 0.5294 |
| [50–60] | 8 | 0.1569 | 0.6863 |
| [60–70] | 12 | 0.2353 | 0.9216 |
| [70–80] | 0 | 0 | 0.9216 |
| [80–90] | 2 | 0.0392 | 0.9608 |
| [90,100] | 2 | 0.0392 | 1 |

*4.8. RQ8. What Is the Correlation of R&D Project Amount Funding with the Number of Publications?*

Table 7 presents the correlational analysis and the estimation parameters of the linear regression between funding and the number of publications. The data indicate a moderate correlation between both variables (i.e., Pearson correlation is equal to 0.584), in which 34.1% of the model is explained by the independent variable (i.e., amount funding). The graphical estimation of the linear relationship between the two variables is shown in Figure 7. It is noteworthy that most observations are close to the linear model, except for the "ACTRIS-2" project, in which funding of slightly less than EUR 10 million gave rise to 441 scientific artifacts.

**Table 7.** Correlation between amount funding and number of publications.

| | **Model Summary** | | | **Parameter Estimates** | |
|---|---|---|---|---|---|
| **Pearson** | **R Square** | **F** | **Sig.** | **Constant** | **Variable** |
| 0.584 | 0.341 | 58.015 | $<1 \times 10^{-3}$ | $-5.751$ | $9.125 \times 10^{-6}$ |

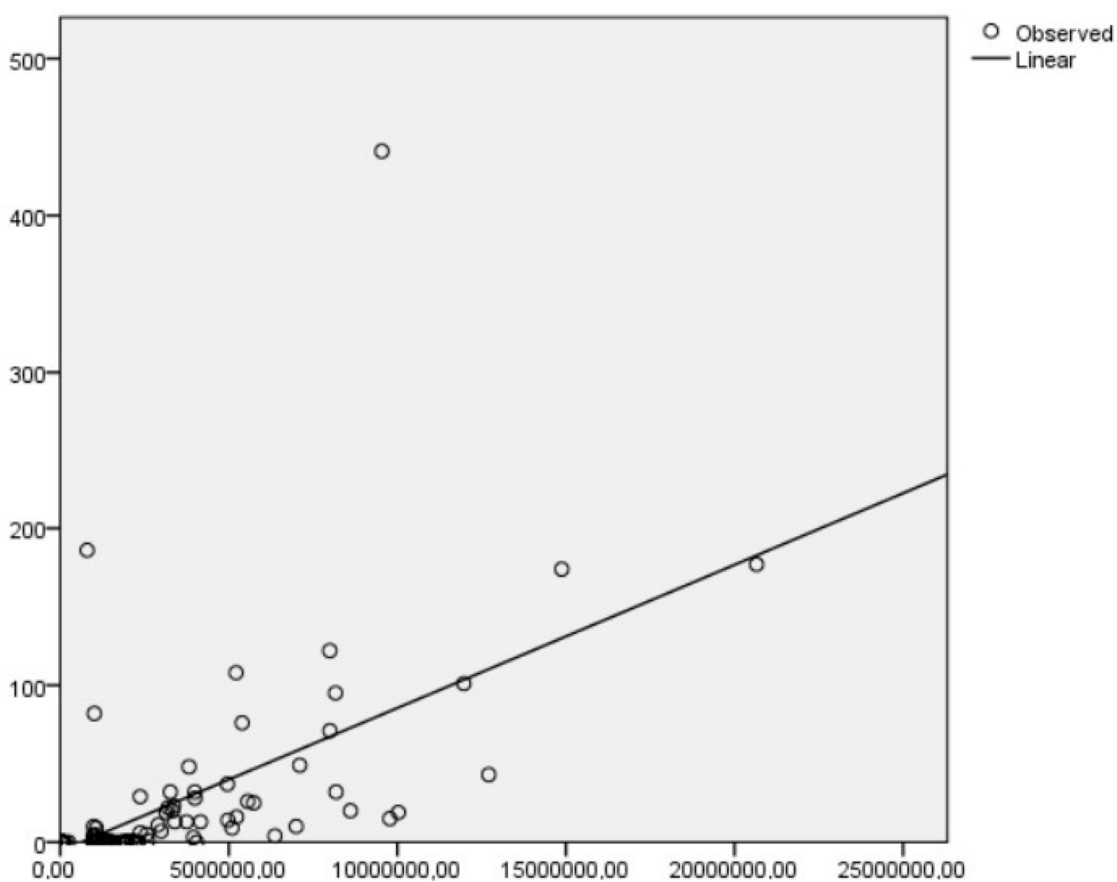

**Figure 7.** Linear regression model between funding and number of publications.

*4.9. RQ9. What Is the Correlation of the Number of Partners in the R&D Project with the Number of Publications?*

Table 8 presents the correlational analysis and the estimation parameters of the linear regression between the number of partners and the number of publications. The data suggest a strong correlation between both variables (i.e., Pearson correlation is equal to 0.760), in which more than 50% of the model (i.e., 57.7%) is explained by the independent variable (i.e., number of partners). The graphical representation of the linear relationship between the two variables is shown in Figure 8. Most observations show a good fit with the linear estimation, except for "ACTRIS-2" because the 441 scientific artifacts of the project

were obtained with the participation of only 50 partners when there are projects with more partners (e.g., "AtlantOS" or "ECOPOTENTIAL") that had less scientific output.

**Table 8.** Correlation between the number of partners and number of publications.

| Model Summary | | | | Parameter Estimates | |
|---|---|---|---|---|---|
| **Pearson** | **R Square** | **F** | **Sig.** | **Constant** | **Variable** |
| 0.760 | 0.577 | 152.938 | $<1 \times 10^{-3}$ | −8.055 | 3.897 |

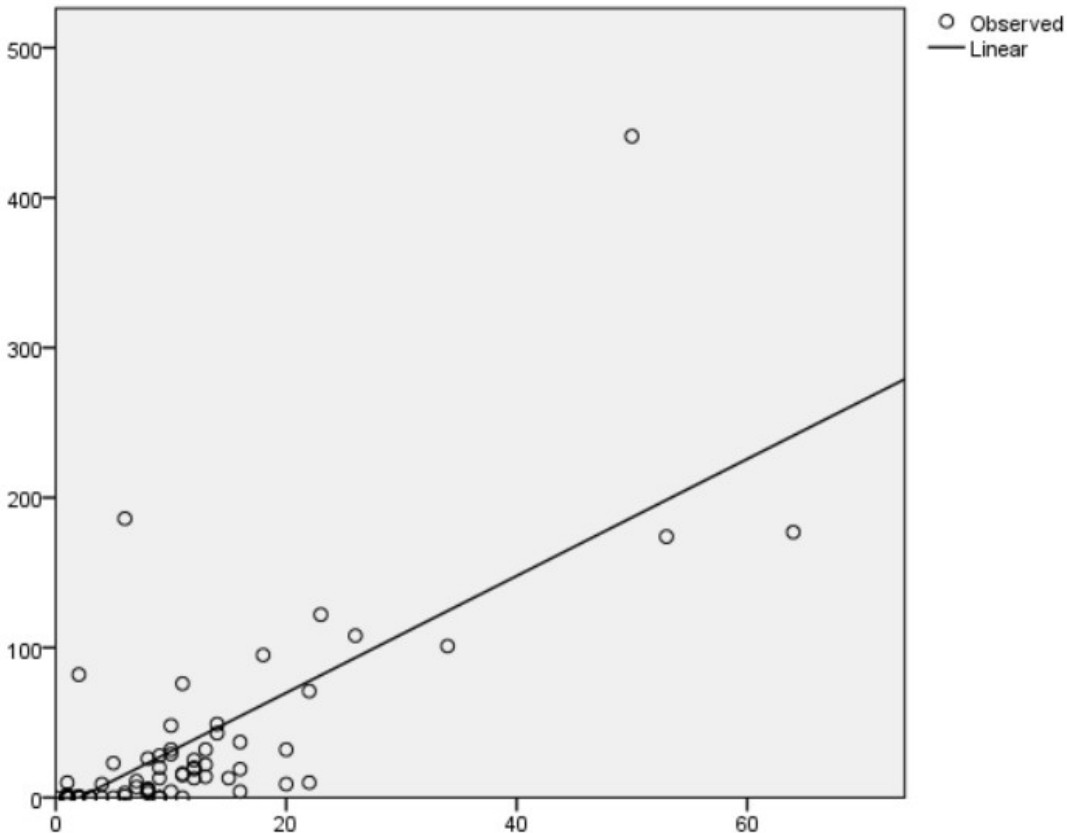

**Figure 8.** Linear regression model between the number of partners and number of publications.

## 5. Discussion

The outcomes of projects funded by Horizon 2020 tend to be quite diverse, although the weight of scientific production (e.g., journals, conference proceedings, book chapters) is approximately 62% higher than the deliverables (e.g., reports, websites, prototypes). These results indicate that the projects simultaneously seek to address the challenge of scientific production and technology transfer. Scientific production assumes the objective of disseminating research to the community, which allows others to use and evaluate it from other perspectives. It is through this medium that the scientific community learns about the results of a research work and what it represents to the collectivity. Smit and Hessels [63] conclude that the publication of a scientific article, in its various forms, is a way to transmit to the technical scientific community the knowledge of new discoveries, and the development of new materials, techniques and methods of analysis in the various areas of science with a potentially high social impact, while technology transfer represents the transfer of know-how, technical knowledge, or technology from one organization to another. In the case of the projects incorporated in this study this transfer is mainly carried out from academia and research centers to the business world directly or through innovation networks, as advocated by Gay [64]. Technology transfer is a process composed of several

steps that are not always incorporated in the scope of the projects or that are finalized at the conclusion of the process, but in this process, there are other activities, such as invention disclosure, patenting, licensing, or royalty management among the shareholders (e.g., universities, R&D centers, or private/public companies) [65].

Communication is an essential activity for the progress of science and the projects funded by Horizon 2020 demonstrate its relevance by evaluating the different types of outcomes of the projects. Abraham [66] highlights that communication is an activity that should be assumed by researchers as central to science, because it puts into motion everything that is vital for research, that is, legitimation and recognition, which will guarantee support and financial resources to researchers. Communication is primordial to scientific advance, this fact is a consensus, since it is through it that it becomes possible to exchange information and ideas between individuals for the feedback of the scientific process. This vision is demonstrated in the vision of Jucan and Jucan [67], which highlight the role of scientific communication in encouraging thinking and action, by insertion or interaction with other people's ideas, knowledge, experience, and accomplishments. It is also important to highlight that every scientist aims for consensus; they want their work to be known by their peers and for them to be convinced of their point of view. The main goals of the exchange of knowledge and information between scientists are to provide answers to specific questions, help scientists keep up to date with new discoveries, help researchers find information about a new scientific field, show the main trends in their field and give importance to their work.

The digitalization of the economy in SMEs and their digital transition represents an opportunity to improve productivity levels, enhance innovation and reduce the costs of business processes. All businesses feel the need to reinvent themselves, analyze, optimize, improve their efficiency and processes and reduce costs, especially in the current context of globalization and economic uncertainty. Empirical studies by Teng et al. [68] and Ulas [69] recognize that the most digital SMEs are the ones that are best prepared for the challenges, and technological solutions are a valuable tool for any SME in any sector, since they allow decision makers to have a complete view of the business, optimizing and streamlining processes, reducing costs and even anticipating possible scenarios, making the best decisions based on credible and reliable insights. This is a theme that throughout the Horizon 2020 program period was addressed in 114 projects, which represents a high concern of the European Commission in gathering initiatives that help SMEs address this challenge. The activity sectors addressed by these projects are quite diverse, which indicates that their results can be applied to a diverse set of industries. However, its applicability in the sustainability area stands out. The evolution of digitalization processes, which have accelerated significantly in recent years due to the demands created by COVID-19 and policy changes of dematerialization of economic activities, as recognized in Amankwah-Amoah et al. [70] and Małkowska et al. [71], has brought the digitalization path in line with sustainability principles. Moreover, the climate emergency in which we find ourselves has made it unavoidable to improve energy efficiency [72]. Also with high relevance are the information and communication, human health, education and construction sectors. Environmental issues are on the priority agenda of the European Commission due to the need to ensure the protection of the environment. This is a theme that has been increasingly addressed by the scientific community, in which the search for balance between the availability of existing natural resources and their exploitation by society is sought, to allow the current generation to develop and, at the same time, to guarantee the next generations the opportunity to also have the same resources for their survival [73–75]. However, sustainability is not only an environmental issue, it is a concept that from the academic point of view, from research funding agendas, and from the connection of research to society, implies a reconceptualization of educational research and the emergence of a new paradigm, marked by a broader, more humanistic and less functional concern for its future. Thus, exploring the theme of sustainability demands from the community a deep reflection on new ways of conceptualizing education and research and the emergence of a

new paradigm for educational research oriented toward the goals of sustainability [76–78]. Furthermore, digital transformation is essentially a social revolution that has changed the way we live, work, study and relate to each other [79]. The challenge is to ensure that the changes that have been introduced by the increasing pace of digitalization can contribute to a more efficient and sustainable society.

The scientific production resulting from projects under Horizon 2020 typically involves several authors from the partner institutions of the projects. Also relevant is the weight of publications that are open access, which represents more than 40% of the scientific production in half of the projects in the sample of this study. Open access publications are an integral part of the policy regarding free access to European-funded projects so that the results of the projects cannot be exclusively exploited by their participants and can benefit society in general. Several arguments can be found in the literature regarding the advantages of open access publishing, such as increased visibility, improved impact of research results and improved monitoring, evaluation and management of scientific activity [80,81]. Moreover, open access scientific production facilitates faster dissemination of knowledge to the entire academic community by increasing the number of citations [82].

Finally, the study addressed the correlation between the amount of project funding, the number of partners, and the number of publications. This correlation is moderate in the first scenario and strong in the second scenario, which indicates that scientific production is mostly fostered in projects with a larger number of partners. The networking of these projects, even if often thematic and scientific subgroups of interest are created, leads to a higher scientific production. In addition, funding programs tend to bring together researchers with different cultures and geographies but who somehow work in common or interdisciplinary areas. Thus, it is recognized that cognitive proximity is a factor that contributes to increased scientific production. Cognitive proximity is addressed in Hautala [83] as an element that gives us information about the similarity by which two researchers share the same knowledge base. Its importance consists in the idea that the effectiveness of knowledge transfer in a collaboration requires a certain level of absorptive capacity for the identification, processing, understanding, interpretation and exploitation of new knowledge. Therefore, it is necessary that the cognitive bases of the researchers be close enough for successful communication and understanding by both parties, so that cognitive proximity between researchers functions as an essential prerequisite for the interactive process of learning and collaboration between them. Cognitive proximity is not only reflected in the context of scientific production but is also an important element for the construction of innovation networks and knowledge sharing in organizations [84,85].

## 6. Conclusions

Digital transformation is a challenge for organizations to work on a global scale and meet the challenges of technological change and new needs and ways of interacting with customers and employees. This is a movement that is not only directed at large corporations but also at SMEs. The entire value chain is undergoing a digital transition, be it the supply chain or the internal and business processes, and SMEs will have to look at innovation. Collaborative networks become fundamental to enable the transfer of scientific and technological knowledge to the business fabric. The European Commission through its support programs for research and innovation takes on the challenge of helping SMEs in this digital transformation process by building collaborative networks. The outcomes of these R&D projects are essentially analyzed from the perspective of deliverables (e.g., prototypes, demonstrators, reports, websites, patents) and scientific production (peer review articles, conference proceedings, book chapters). The results of this study allowed us to conclude that both types of outcomes are important for the projects, although there is a greater weight for the scientific component, which represents more than 60% of these outcomes.

The projects funded under Horizon 2020 have an inter-sectoral approach, essentially focusing on projects related to the theme of multidisciplinary sustainability in which the specific impact on sectors of activity, such as water supply, information and communication,

human health, education and construction, stand out. This situation allows us to conclude that sustainability is a theme that has multiple perspectives, with the environmental component being only one of them. Horizon 2020 has contributed to a high international collaboration with about 70% of the scientific production involving researchers from several partner institutions in the projects. Scientific production in open access is not yet a priority for all projects and there is a high level of heterogeneity between projects. The correlation between the number of project partners and the number of publications is strong, while the correlation is only moderate when we consider the relationship between the amount of funding obtained and the number of publications.

This work offers both theoretical and practical contributions. From a conceptual perspective, this study allows us to explore in relative depth the characterization of the scientific production carried out within projects funded by Horizon 2020 in the digital transformation of SMEs. The scientific production of these projects was explored considering multiple perspectives, such as the type of publication, sectors of activity addressed, percentage of international collaboration, percentage of publications in open access and the relationship between the funding obtained and the number of partners with the amount of scientific production. From a practical perspective, the results of this study can be used by both European and national funding entities and SMEs. European or national funders can better direct their financial support to projects whose outcomes are more aligned with their objectives and improve the performance of these projects. We highlight the importance that funds supporting scientific research can play in bringing together the business sector (particularly SMEs) and universities, helping to implement the third mission of universities related to the transfer of technology and knowledge. The projects in which businesses and universities participate present better conditions for scientific evidence, both in terms of technical deliberations and scientific production. Furthermore, this financial support may encourage the emergence of innovation consortia in the future and improve the competitiveness of SMEs. Additionally, SMEs can use these results to improve the structuring of their project proposals, and thus obtain better chances of obtaining funding. Finally, it is also relevant to explain the main limitations of this study. The analysis of scientific production was accounted for exclusively from a quantitative perspective without analyzing the quality of this scientific production, namely in terms of its impact considering the main scientific indexers, such as Web of Science or Scopus. Therefore, and as future work, it would be desirable to extend this analysis considering whether the projects with the highest scientific production are also those that present the highest quality of that production. Another limitation is related to the methodological process that looks at the outcomes of the project without considering its strategic objectives. For example, the project "Con-struye2020_Plus" presents very low levels of collaboration in scientific production (less than 3%) but which are justifiable given its scientific production strategy. The same observation can be made concerning the scientific production since the alternatives and costs of scientific production depend strongly on the areas of each project and the typology of this scientific production. In this sense, and as future work, it would be desirable to conduct a study to conclude whether scientific production in open access has allowed an increase in the international impact of these publications, considering multiple perspectives, such as the impact factor of these journals, indexing databases and number of citations, among others.

**Author Contributions:** Conceptualization, F.A., J.M. and J.D.S.; methodology, F.A.; validation, F.A., J.M. and J.D.S.; formal analysis, F.A.; investigation, F.A., J.M. and J.D.S. writing—original draft preparation, F.A.; writing—review and editing, J.M. and J.D.S. All authors have read and agreed to the published version of the manuscript.

**Funding:** This research received no external funding.

**Data Availability Statement:** Data are available at CORDIS website: https://cordis.europa.eu (accessed on 30 June 2022).

**Conflicts of Interest:** The authors declare no conflict of interest.

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
