# Peer review of "A Bibliometric Analysis of the Scientific Outcomes of European Projects on the Digital Transformation of SMEs"

_publications, doi:10.3390/publications10040034_

Round 1

Reviewer 1 Report

In general, this article is written on a very relevant topic. It is useful not only from the point of view of increasing scientific knowledge in the field of science, but also from a practical point of view, since its results can be useful for making managerial decisions both directly by small and medium enterprises themselves and by funds that finance scientific research. 

It is especially necessary to note the author's approach to the research methodology, which is presented very clearly and can be used by any scientists to conduct research of this kind in the framework of studying the effectiveness of funded projects.

An important advantage of this article is also the consistent formulation of research questions and their disclosure on the basis of the research.

Author Response

We appreciate the review suggestions and comments received by the reviewer. These elements are key to improving the final quality of the manuscript. Below we respond to each issue raised.

Review #1

In general, this article is written on a very relevant topic. It is useful not only from the point of view of increasing scientific knowledge in the field of science, but also from a practical point of view, since its results can be useful for making managerial decisions both directly by small and medium enterprises themselves and by funds that finance scientific research.

Author’s response: Thank you very much regarding the implications and relevance of this study for SMEs and founding agencies that support the development of scientific research. We also believe that it is important to offer both theoretical and practical implications which are addressed in the Conclusions section.

It is especially necessary to note the author's approach to the research methodology, which is presented very clearly and can be used by any scientists to conduct research of this kind in the framework of studying the effectiveness of funded projects.

Author’s response: Thank you very much for your feedback regarding the presentation of the research methodology.

An important advantage of this article is also the consistent formulation of research questions and their disclosure on the basis of the research.

Author’s response: Thank you very much for your positive evaluation about the relevance of the research questions and their exploration in the Results section.

Reviewer 2 Report

I have not been able to trace the data contained in figure 3 versus the quantities in figure 4, according to figure 3 we have 2,348 publications per year, I am not able to relate with the quantities shown in figure 4.

It is possible to have incorporated bibliometric analyses that would have made the research more robust, such as the most productive authors, or the most cited authors, or the most cited publications, or to have generated clusters with the different associations of authors.

Author Response

We appreciate the review suggestions and comments received by the reviewer. These elements are key to improving the final quality of the manuscript. Below we respond to each issue raised.

Review #2

I have not been able to trace the data contained in figure 3 versus the quantities in figure 4, according to figure 3 we have 2,348 publications per year, I am not able to relate with the quantities shown in figure 4.

Author’s response: Thanks for your observation and opportunity to clarify this issue. The data were correct but the legend of figure 3 was incorrect. Therefore, we have replaced the Figure 3 to distinguish between other deliverables and other publications. We have also clarified the total number of deliverables and total number of publications in the text. The projects present a total of 1460 deliverables and 2348 publications.

It is possible to have incorporated bibliometric analyses that would have made the research more robust, such as the most productive authors, or the most cited authors, or the most cited publications, or to have generated clusters with the different associations of authors.

Author’s response: Thank you very much for your suggestion. We have followed your recommendation and included a new research question to explore the most cited authors and documents. Consequently, we have also improved the Results section and included two new tables: Table 2 to present the top-10 most-cited authors and Table 3 to present the top-10 most-cited documents. Furthermore, we have included two new references to support this research question:

Hausberg, J.P.; Liere-Netheler, K.; Packmohr, S.; Pakura, S.; Vogelsang, K. Research streams on digital transformation from a holistic business perspective: a systematic literature review and citation network analysis. J. Bus. Econ. 2019, 89, 931-963. https://doi.org/10.1007/s11573-019-00956-z

Hanelt, A.; Bohnsack, R.; Marz, D.; Marante, C.A. A Systematic Review of the Literature on Digital Transformation: Insights and Implications for Strategy and Organizational Change. J. Manag. Stud. 2021, 58, 1159-1197. https://doi.org/10.1111/joms.12639

Reviewer 3 Report

The bibliometric study has its merits. We do learn from it and the way questions and methodology together with results is described seem coherent.

One session where i see most space for improvement is related to RQ4 and section 4.4. I think this question has potentially wider interest and could be both explained and addressed more in depth. What we learn from these conclusions seem quite superficial while the question when addressed more in depth could provide the most relevant implications from this kind of study.

Similarly, if i refer to line 457 and the argument that starts there, i find that studies like this one should be particularly relevant to enhance the possibility to make good evaluations (like the ones suggested in that section) by entities like not only SMEs but most importantly Europearn and national funding entities. While the study suggests in the discussion that results from this study can be used for different goals, what would be mostly appreciated in a study like this is to have a sharp clear description of HOW exactly such entitites can use the results proposed in this study to reach such goals while the discussion remains vague and quite abstract and high level.

If you could have a full section in the paper where you more clearly explain the HOWs, that would be awesome and make the study really relevant. I believe otherwise, although correct, it remains quite high level and its difficult to appreciate its implications.

Best of luck to the authors and thank you for the interesting reading :-)

Author Response

We appreciate the review suggestions and comments received by the reviewer. These elements are key to improving the final quality of the manuscript. Below we respond to each issue raised.

Review #3

The bibliometric study has its merits. We do learn from it and the way questions and methodology together with results is described seem coherent.

Author’s response: Thank you very much for your positive feedback regarding the relevance and coherence of this study.

One session where i see most space for improvement is related to RQ4 and section 4.4. I think this question has potentially wider interest and could be both explained and addressed more in depth. What we learn from these conclusions seem quite superficial while the question when addressed more in depth could provide the most relevant implications from this kind of study.

Author’s response: Thanks for your suggestion. We also agree this research question is very relevant and could be better explored. Firstly, we would like to inform that we have included a new research question related to the most cited authors and documents and, therefore, this research question is now coded as RQ5. We have improved the discussion between the relevance of the digitalization for sustainability and explained how these two concepts are interconnected. Accordingly, we have included four new references:

Amankwah-Amoah, J.; Khan, Z.; Wood, G.; Knight, G. COVID-19 and digitalization: The great acceleration. J. Bus. Res. 2021, 136, 602–611. https://doi.org/10.1016/j.jbusres.2021.08.011

Malkowska, A.; Urbaniec, M.; Kosala, M. The impact of digital transformation on European countries: insights from a com-parative analysis. Equil. 2021, 16, 325-355. http://dx.doi.org/10.24136/eq.2021.012

Taddeo, M.; Tsamados, A.; Cowls, J.; Floridi, L. Artificial intelligence and the climate emergency: Opportunities, challenges, and recommendations. One Earth 2021, 4, 776-779. https://doi.org/10.1016/j.oneear.2021.05.018

Knell, M. The digital revolution and digitalized network society. Rev. Evol. Polit. Econ. 2021, 2, 9-25. https://doi.org/10.1007/s43253-021-00037-4

Similarly, if i refer to line 457 and the argument that starts there, i find that studies like this one should be particularly relevant to enhance the possibility to make good evaluations (like the ones suggested in that section) by entities like not only SMEs but most importantly Europearn and national funding entities. While the study suggests in the discussion that results from this study can be used for different goals, what would be mostly appreciated in a study like this is to have a sharp clear description of HOW exactly such entitites can use the results proposed in this study to reach such goals while the discussion remains vague and quite abstract and high level.

If you could have a full section in the paper where you more clearly explain the HOWs, that would be awesome and make the study really relevant. I believe otherwise, although correct, it remains quite high level and its difficult to appreciate its implications.

Best of luck to the authors and thank you for the interesting reading :-)

Author’s response: It is a very pertinent suggestion and we have addressed it in the Conclusions section. It is relevant to explain how European and national funding entities can use the results of this study to offer more robust support policies that may have more relevant social impacts considering the various actors involved in this process. We highlight the importance that funds supporting scientific research can play in bringing together the business sector (particularly SMEs) and universities, helping to implement the third mission of universities related to the transfer of technology and knowledge. The projects in which business and universities participate present better conditions for scientific evidence both in terms of technical deliberations and scientific production. Furthermore, this financial support may encourage the emergence of innovation consortia in the future and improve the competitiveness of SMEs.